# Real-World Effectiveness of COVID-19 Vaccine and Identification of SARS-CoV-2 Variants among People Living with HIV on Highly Active Antiretroviral Therapy in Central Kerala of India—An Ambi-Directional Cohort Study

**DOI:** 10.3390/v15112187

**Published:** 2023-10-30

**Authors:** Joe Thomas, Priyanka Rajmohan, Ponnu Jose, Radhika Kannan, Rosmi Jose, Unnikrishnan Uttumadathil Gopinathan, Lucy Raphael, Nithya M. Baiju, Swathi Krishna, Teny Attokaran, Jubina Bency A. T, Aiswarya Venugopal, Soorya Sheela, Akhila Kallempadam, Lee Jose, Susheela J. Innah, Pulikkottil Raphael Varghese, Alex George

**Affiliations:** 1Department of Community Medicine, Jubilee Mission Medical College and Research Institute, Thrissur 680005, Kerala, India; covidjmmcri@gmail.com (J.T.); priyankarajmohan@gmail.com (P.R.); ponnu034@gmail.com (P.J.); unni.thsr@gmail.com (U.U.G.); drlucyraphael@gmail.com (L.R.); tenytharu@gmail.com (T.A.); aiswaryavenu90@gmail.com (A.V.); soorya3427@gmail.com (S.S.); k.akhila13@gmail.com (A.K.); leejosesr19@gmail.com (L.J.); 2Department of Microbiology, Jubilee Mission Medical College and Research Institute, Thrissur 680005, Kerala, India; rosdinmi@gmail.com; 3Department of Transfusion Medicine, Jubilee Mission Medical College and Research Institute, Thrissur 680005, Kerala, India; nithyambaiju1988@gmail.com (N.M.B.); innah2k@gmail.com (S.J.I.); 4KEM Hospital Research Centre, Pune 412216, Maharashtra, India; swathikdk@gmail.com; 5Department of Community Medicine, PK DAS Medical College, Vaniyamkulam 679522, Kerala, India; jubina.bency@gmail.com; 6Jubilee Centre for Medical Research, Jubilee Mission Medical College and Research Institute, Thrissur 680005, Kerala, India; drprvarghese@gmail.com (P.R.V.); alexgeorge@jmmc.ac.in (A.G.)

**Keywords:** SARS-CoV-2 variants, COVID-19 vaccine, HIV/AIDS, ChAdOx1 COVID-19 vaccine, CD4 cell counts, BBV152 COVID-19 vaccine

## Abstract

Background: Vaccine effectiveness for first-generation coronavirus disease (COVID-19) vaccines among People Living with HIV (PLHIV) in India remains unexplored. This study entails the estimation of the real-world effectiveness of COVID-19 vaccines (AZD1222/Covishield, BBV152/Covaxin) among PLHIV and the identification of variants of SARS-CoV-2 among those infected with COVID-19. Methods: An ambi-directional cohort study was conducted among 925 PLHIV above 18 years of age in two districts of central Kerala, India, from February 2022 to March 2023. Selected PLHIV were recruited as Participant Liaison Officers (PLOs) for the follow-up on the study participants. At enrolment, basic details, baseline CD4 count, and a Nasopharyngeal (NP) swab for RT-PCR were collected. In the follow-up phase, NP swabs were collected from subjects with COVID-19 symptoms. Positive subjects had a CD4 count and genomic sequencing performed. Results: The mean age of the participants was 46.93 ± 11.00 years. The majority, 819 (93.6%), of participants had received at least one dose of any vaccine, while 56 (6.4%) were unvaccinated. A total of 649 (79.24%) participants were vaccinated with Covishield and 169 (20.63%) with Covaxin. In the vaccinated group, 158 (19.3%) reported COVID-19 infection. Vaccine Effectiveness (VE) for one dose of any vaccine was 43.2% (95% CI: 11.8–64.5), *p* = 0.015. The effectiveness of full vaccination with Covishied was 63.8% (95% CI: 39.3–79.2), *p* < 0.001, and Covaxin was 73.4% (95% CI: 44.3–87.3). VE was highest, at 60.7% (95% CI: 23.6–81.3), when the two doses of the vaccine were given at an interval of less than 6 weeks. Participants with a baseline CD4 count > 350 had greater protection from COVID-19, at 53.4% (95% CI: 19.6–75.3) *p* = 0.004. The incident cases were sub-variants of Omicron (BA.2, BA.2.38, BA.2.10). Conclusions: Full vaccination with Covishield and Covaxin was effective against COVID-19 infection among PLHIV on treatment; albeit, that of Covaxin was higher. A gap of 4 to 6 weeks between the two doses of COVID-19 vaccine was found to have higher VE among PLHIV.

## 1. Introduction

The WHO announced coronavirus disease (COVID-19) as a global pandemic in the month of March 2020. On 24 November 2021, a new SARS-CoV-2 variant was reported to the WHO from South Africa which was officially named Omicron (B.1.1.529), as a variant of concern. The rapid spread of this highly mutated strain across 50 countries including India in less than a week sparked a global health alarm. The mutations in the Omicron variant have indicated increased transmissibility and vaccine resistance due to its immune escape phenomenon [1,2]. Kerala was in the receding phase of the third wave of the pandemic during this study, with 38,400 active cases, 1.3% of them having severe COVID-19 [3].

Emergency use of two COVID-19 vaccines, viz., Covishield and Covaxin, was introduced by the Government of India on 16 January 2021. AZD1222/Covishield (manufactured by Serum Institute of India Limited) is a recombinant, replication-deficient chimpanzee adenovirus vector encoding the SARS-CoV-2 Spike (S) glycoprotein produced in genetically modified human embryonic kidney (HEK) 293 cells. BBV152/Covaxin (Manufactured by Bharat Biotech Limited, India) was developed using the whole-Virion SARS-CoV-2 vaccine strain NIV-2020-770, utilising inactivated Vero Cell-derived platform technology. Both vaccines were initially given as two doses at 4 week intervals [4,5]. However, for the AZD1222/Covishield vaccine, the interval between two doses was increased to 84 days by June 2021 by the Ministry of Health and Family Welfare for adults aged above 18 years. During the initial roll-out of the vaccines, healthcare workers, individuals above the age of 60 years, and those with comorbidities were provided vaccinations. Reports have shown that during this period, the vaccination coverage among People Living with HIV/AIDS (PLHIV) was lower than that of the general population [6].

India has the third-largest burden of Human Immunodeficiency Virus (HIV) infection in the world, the estimated numbers being 2.4 million, and that of Kerala was 24,481 in 2021 [7]. PLHIV are a vulnerable group, and their vaccination needs to be prioritized during a pandemic [8]. Studies conducted in many regions around the world suggest varying immune responses to COVID-19 infection among vaccinated PLHIV, suggesting a different degree of viral control and immune reconstitution of PLHIV [9,10,11]. Presently, no study has been published on the Vaccine Effectiveness (VE) of COVID-19 vaccines among PLHIV in India. This study entails the estimation of the real-world effectiveness of COVID-19 vaccines among PLHIV and the identification of variants of the SARS-CoV-2 virus in PLHIV who are infected with COVID-19. 

## 2. Materials and Methods

An ambi-directional cohort study was conducted among the members of PLHIV support groups in 2 districts of central Kerala (Thrissur and Palakkad) in India, from February 2022 to March 2023. We included all PLHIV above the age of 18 years who consented to participate and excluded participants who were unable to communicate coherently or were non-ambulatory. Using the Slovins formula: {n = N/(1 + Ne^2^)}, accounting for 15% attrition and 3% margin, the calculated sample size was 925. A line list of all the members of the PLHIV support groups of the two districts was collected and the required sample was selected using the simple random sampling method. 

In order to facilitate social networking and to overcome the barriers in communication between the investigators and the study participants, five PLHIV who had the aptitude for communication and dissemination of information were recruited from the HIV/AIDS care and support groups of the Thrissur and Palakkad districts as Participant Liaison Officers (PLO) to follow-up and monitor the study participants during the prospective phase. A two day intensive training was conducted at our institution to sensitise the PLOs about the epidemiology of COVID-19, COVID vaccines, and the study protocol. A study manual was prepared and given to the PLOs to understand the guidelines and the standard operating procedures. A supervisor (health care worker) monitored the work performed by the PLOs and followed up the COVID-19 positive subjects. A Quality Assurance (QA) manager from the central team periodically verified the source register data and validated the follow-up visits conducted by the PLOs by randomly cross-checking 10% of participants. Regular QA reports were filed by the QA manager and corrective measures, if required, were taken by the investigators.

### 2.1. Enrolment Process

After obtaining written informed consent, a study identity card was issued, bearing the subject ID of the participant and the contact number of PLO. In order to maintain confidentiality and anonymity for all future references, the participant was referred to using the subject ID number. The study participants were interviewed regarding socio-demographic details, latest CD4 count, viral load, ART status, prior COVID-19 infection and vaccination history. The vaccination status was confirmed from the government vaccination portal (Cowin portal) or by checking their vaccination certificates. Their baseline blood sample was collected for CD4 count estimation and nasopharyngeal (NP) swab was taken for RT-PCR testing (Figure 1).

### 2.2. Follow-Up Process

Each PLO was assigned 180 participants for follow-up. Vaccination status of each participant was updated by the PLOs when they were vaccinated against COVID-19. The participants were contacted by the PLO (30 participants/day) on a weekly basis to enquire for any symptoms suggestive of COVID-19 infection. Participants also self-reported symptoms to the PLO by calling or messaging into the dedicated phone number written on the study ID card. Symptomatic subjects were directed to our institution or to the nearest link centres for testing (ICMR approved RT-PCR) and treatment. Any laboratory confirmed SARS-CoV-2 infection among the participants were immediately reported to the supervisor by the PLO. The mobile lab team consisting of supervisor and phlebotomist collected blood samples for CD4 counts and NP swabs from the participant within 24 h. The swabs were transported in viral cold-chain transport medium to our institution for genomic sequencing. The SARS-CoV-2 viral amplification was conducted employing the Mid Night Protocol and sequencing was performed using illumina platform. The samples of study participants who tested positive for COVID-19 were subjected for whole genome sequencing with a high CT Value (CT value 20–25). The high-quality reads of the samples were aligned to the severe acute respiratory syndrome coronavirus 2 isolate Wuhan-Hu-1, complete genome (NC_045512.2) reference sequence using BWA MEM (version 0.7.17). Consensus sequence was extracted using Samtools mpileup. The mpileup utility of Samtools (v 1.9) was used to identify SNPs from the sorted BAM file of the samples. The SNPs were filtered based on a minimum read depth of 5, quality threshold of 25.

### 2.3. Ethical Considerations

The study was registered in Clinical Trial Registry of India (No: CTRI/2021/10/037337) and approvals from the Institutional Ethics Committee (Ref.No:79/21/IEC/JMMC&RI) and Kerala State AIDS Control Society (KSACS) were obtained before initiation of the study. At enrolment, participant informant sheets were explained in vernacular language to each participant. Only those participants who gave written informed consent were enrolled into the study. A copy of the participant information sheet and consent form was given to the participant.

Data were analysed using IBM SPSS version 25.0. Categorical variables were represented as frequencies and percentages and quantitative variables as mean ± standard deviation or median (IQR). In the primary analysis, fully vaccinated individuals were compared with unvaccinated. Secondary analyses were conducted for partially vaccinated and targeted subgroups. Multivariable logistic regression model adjusted for age, gender, and employment was used to find vaccine effectiveness [VE = {1 − (aRR)} × 100%].

## 3. Results

### 3.1. Socio-Demographic Distribution

We recruited 925 PLHIV subjects and the mean age of the participants was 46.93 ± 11.00 years with the majority, 358 (38.7%), in the age group of 41 to 50 years. (Table 1) There were 500 (54.1%) males, 424 (45.8%) females, and one (0.5%) transgender in our study. A higher number of subjects, 583, were married (63%), 684 had completed a secondary level of education (73.9%), and 652 were employed (70.5%).

### 3.2. Clinical Characteristics of PLHIV Subjects

The mean duration of HIV infection in the study participants was 10.51 ± 5.60 years and the mean duration of treatment was 8.48 ± 4.75 years. The majority, 767 (82.9%), of PLHIV subjects were treated with the TLD (Tenofovir, Lamivudine, and Dolutegravir) regime.

Among the study participants, 356 (38.5%) had reported prior history of at least one comorbidity. The most common comorbidity was diabetes mellitus, 141 (15.2%), followed by hypertension, 78 (8.43%), and CAD, 31 (3.4%). A past history of stroke was reported by 13 (1.4%). A history of opportunistic infections (OI) was reported by 274 (29.6%) participants. The most common OI was tuberculosis, 220 (23.8%), and candidiasis, 36 (3.9%).

Among the 925 subjects, 194 (20.97%) had reported COVID-19 infection in the retrospective phase of our study and 10 (1.08%) cases were identified during the follow-up period. None of them reported severe COVID-19 infection or hospitalisation during our study period. In patients with pre-existing comorbidities, COVID-19 was reported in 90 (26%) participants. Out of 204 (22.05%) participants with COVID-19 infection, 69 (25.2%) had OIs.

### 3.3. Association between Baseline Demographic and Clinical Characteristics with COVID-19

On analysing the association between sociodemographic characteristics such as age, gender, marital status, education, and occupation with COVID-19 infection, there was no significant association with the variables studied.

Those with comorbidities had a significantly higher risk of developing COVID-19 compared to those without comorbidities; aRR = 1.32 (95% CI: 1.04–1.68); *p* = 0.025. Among the different types of comorbidities, those with diabetes were found to have a significantly higher risk of COVID-19, aRR = 1.44 (95% CI: 1.08–1.92); *p* = 0.016. Participants with a history of opportunistic infection had a higher risk of developing COVID-19 than others, though this was not statistically significant (Table 2). 

### 3.4. CD4 Count and Viral Load among PLHIV

The median CD4+ T lymphocyte count of the HIV participants was 523 (IQR:371–729) cells/mm^3^. Out of 194 (22.42%) participants with a low CD4 count (<350 cells/mm^3^), 36 (18.6%) subjects reported COVID-19 infection. Among those with a CD4 > 350 cells/mm^3^, 138(20.6%) had reported infection with COVID-19. The median CD4 count among those infected with COVID-19 was 557 (IQR: 386–734) and the count among those PLHIV subjects not infected with COVID-19 was 516 (IQR: 366.75–728). But, this difference was not found to be statistically significant.

Among 701 participants whose viral load test result was available, the majority, 666 (75.03%), had viral load suppression (<1000 copies/mL) in the past year; among them, 140 (21%) had COVID-19 infection. Among those with unsuppressed viral load (>1000 copies/mL), four (11.4%) had COVID-19 infection. There was no significant association between viral load suppression and infection with COVID-19.

### 3.5. Vaccination Status among PLHIV Subjects

The majority of the subjects, 819 (93.6%), had received at least one dose of any vaccine and 56 (6.4%) were unvaccinated. The vaccination history was confirmed by verifying vaccination certificates, SMS messages, or through the Cowin portal. The majority, 649 (79.24%), had been vaccinated with Covishield, 169 (20.63%) with Covaxin, and 2 (0.24%) with the sputnik vaccine. Only those participants who had taken either Covishield or Covaxin were included in further analysis (n = 818). Among the vaccinated individuals, 650 (79.46%) had been fully vaccinated (completed two doses of the vaccine > 14 days), and 168 (20.53%) had been partially vaccinated (either one dose or two doses < 14 days). Precautionary doses of the vaccine had been taken by 14 (1.711%) participants. Among the vaccinated individuals, 500 (61.12%) had been fully vaccinated with Covishield, 149 (18.21%) had been partially vaccinated, and 8 (0.98%) had taken the precautionary dose of Covishield. Among those vaccinated with Covaxin, 150 (18.33%) were fully vaccinated, 19 (2.32%) partially vaccinated, and 6 (0.73%) had taken the precautionary dose.

### 3.6. Vaccine Effectiveness (VE) by Type, Dose, and Interval between Doses

In the vaccinated group, 158 (19.3%) had reported COVID-19 infection after taking the vaccine, whereas 661 (80.7%) subjects were not infected with COVID-19 after vaccination. The incidence density of COVID-19 among vaccinated individuals was 0.045/100 person days (PD). The incidence density of COVID-19 among individuals fully vaccinated with Covishield was 0.047/100 PD and Covaxin was 0.031/100 PD.

We adjusted for potential confounders such as age, gender, and occupation. The adjusted VE for at least one dose of any vaccine was 43.2% (95% CI: 11.8%, 64.5%) *p* = 0.015. The effectiveness of full vaccination with Covishied was 63.8% (95% CI 39.3%, 79.2% *p* < 0.001) and that of Covaxin was 73.4% (95% CI 44.3%, 87.3%). Partial vaccination with either Covishield, VE = 0(95% CI: 0–4.4) or Covaxin, VE = 0(95% CI 0–49.7%) was found to have no protective effect against COVID-19 (Figure 2).

VE was highest, 60.7% (95% CI: 23.6%, 81.3%), at an interval less than 6 weeks between the two doses and was significantly higher (*p* = 0.004) than the effectiveness during the 6 to 8 weeks interval. VE at the 9 to 11 weeks interval was 59.5% (95% CI: 12.8%, 83.3%; *p* = 0.018). A gap of more than 11 weeks between the two doses showed a VE of 36.5% (95% CI: 2.32–61; *p* = 0.038) (Table 3) (Figure 3).

### 3.7. CD4 Count, Viral Load, Treatment Regime, and VE

We estimated the VE against the CD4 count, setting cut offs at < or =350. The adjusted VE for any COVID-19 vaccine for participants with a CD4 < or =350 was 16.4% (95% CI: 0–67.6). The participants with a CD4 count > 350 had greater protection from COVID-19 infection as compared to those with a CD4 < 350, 53.4% (95% CI: 19.6–75.3) *p* = 0.004. On secondary analysis with Covishield and Covaxin, both the vaccines offered higher protection among those with a higher CD4 count [50.9 (95% CI: 15.8–73.9) *p* = 0.007; 63.1% (95% CI: 28–82.8) *p* = 0.002] (Table 4). In the study group, 21 (2.5%) had a CD4 count less than 100. The estimated VE in this cohort was 76.5% (95% CI: 24–92.7) *p* = 0.022.

On estimation of VE among those subjects on different ART regimes, it was found that VE was greater among those subjects on second-line ART [49.1% (95% CI: 0–74.4) *p* = 0.079] than among those on first-line ART, which was mainly the TLD regime [40.7 (95% CI: 4.6–63.1) *p* = 0.043]. A subgroup analysis was conducted among the subjects with viral load suppression and the estimated VE was 36.8% (95% CI: 0–63.7) *p* = 0.129.

### 3.8. SARS-CoV-2 Variants Infecting PLHIV

A total of 120 Single Nucleotide variants (SNVs) were identified in the ten samples sequenced, where 25% of the SNPs were shared by all and the genes ORF1ab and S were shared by 50% of the SNVs identified (Appendix A: SARS-CoV-2 lineages identified among the samples obtained). The lineages identified were BA.2 (B.1.1.529.2, 40%), BA.2.10 (B.1.1.529.2.10, 20%), BA.2.38 (B.1.1.529.2.38, 20%), and BF.3 (B.1.1.529.5.2.1.3, 20%).

### 3.9. Duration to Development of COVID-19 in PLHIV after Full Vaccination

The median duration to development of COVID-19 in PLHIV after full vaccination was 96.5 days (48.5–157.25) (Figure 4).

## 4. Discussion

Our study to assess the VE among PLHIV indicates that full vaccination with both Covishield and Covaxin showed moderate VE against COVID-19 infection. BBV152/Covaxin had a higher effectiveness, 73.4% (95% CI 44.3%, 87.3%), than AZD1222/Covishield, 63.8% (95% CI 39.3%, 79.2%). The highest reduction in the risk of COVID-19 infection was observed on administering the vaccine at an interval of less than 6 weeks.

Our study group included PLHIV participants recruited from the care and support groups from two districts of Central Kerala with regular follow up at the ART centre with a healthy range of CD4 counts, 523 (IQR: 371–729) cells/mm^3^, and a majority with suppressed plasma viral load. This could be the reason that our VE estimates are comparable to that of the general population. This was consistent with the findings of a clinical trial among PLHIV on ART by Frater J et al., where the vaccine efficacy estimates of AZD1222/Covishield were comparable to healthy adults [12].

Our VE estimates were similar to that in a study by Fowokan A et al., in Canada among PLHIV which showed that the adjusted VE of COVID vaccines 7–59 days after the second dose was 71.1% and increased to 89.3% 60–89 days after the second dose [13]. A study performed by Kuan-Yin Lin et al. in Taiwan demonstrated that COVID-19 vaccination was clinically effective among PLHIV with a VE of 99.9% after two doses of the COVID-19 vaccine [14]. This is likely higher than our estimates because the former study was conducted in a setting with a low endemicity of COVID-19, where non-pharmaceutical interventions were strictly implemented. Our study was conducted during a period when the lockdown measures were relaxed in our country and the real-world performance of the first-generation COVID vaccines was influenced by the newer variants of SARS-CoV-2.

The median CD4+ lymphocyte count of our participants was 523 (IQR: 371–729) cells/mm^3^ and participants with a CD4 count > 350 had greater protection from COVID-19 infection as compared to those with a CD4 count < 350, 53.4% (95% CI: 19.6–75.3). Our results were comparable to study by Carlo Bien’kowski et al. in Poland, where the median CD4+ count was 591 cells/uL (IQR: 459.5–745.0 cells/uL), with fewer breakthrough infections after full vaccination and none of them reported severe disease [11]. Our findings were consistent with those of multiple studies among PLHIV, which demonstrated that seroconversion rate, anti-Spike antibodies, and neutralising titers in PLHIV with CD4 counts over 500/mL were comparable with those in healthy controls [15,16,17,18].

Our VE estimates were comparable to the results from a study by Nittayasoot N et al. in Thailand among the general population, which showed VE ranging from 68.29% to 75.71% for two doses of Covishield [19]. The total VE of complete vaccination with any vaccine was found to be 83% in a study conducted by Tarun Bhatnagar et al. in India. The same study found that the VE of full vaccination with Covishield was 85% and that of Covaxin was 71% [20]. The estimate of overall VE and the VE of Covishield was lower, but the VE of Covaxin was comparable to the results of our study. This difference could be because our study was conducted at a time when the Omicron wave of COVID-19 was ongoing in India. The comparatively lower VE in our study might be due to the immune escape phenomenon shown by the Omicron variant of COVID-19, thereby confirming the existing evidence regarding the lower effectiveness of first-generation COVID vaccines against novel COVID variants. Preliminary vaccine efficacy studies also show that after full vaccination with COVID-19 vaccines, the neutralizing antibody titer against the Omicron variant was low, indicating a below par protection against the Omicron variant [20,21].

Our study found that VE was maximum (60.7%) at an interval of 6 weeks between the two doses. Similar VE estimates were seen in a study by Tarun Bhatnagar et al. on COVID vaccine effectiveness among the general population, which showed 67% effectiveness for Covaxin, whereas for Covishield, the VE was 81%. In their study, VE was highest at an interval of 6–8 weeks between the two doses for AZD1222/Covishield and Covaxin [20].

A molecular characterisation performed among the study participants positive for SARS-CoV-2 demonstrated that all the samples were variants of Omicron and its sub lineage. Our finding of increasing variations in omicron lineage could be due to the rapid mutation of the variant. This could result in an increased risk of breakthrough infections and would reduce the efficiency of immunogenicity elicited by the existing vaccines [22].

None of our participants reported severe COVID-19 infection, hospitalisation, or death due to COVID. This corroborated the findings of a study from New York among PLHIV which reported that HIV infection was not identified as an important comorbid condition in hospitalized patients with COVID-19 [23]. Similar findings were also reported from a study from China which showed that the incidence rate and adverse outcomes of COVID-19 were comparable to the general population [24].

This study estimated the real-world effectiveness of COVID vaccines among PLHIVs on HAART, which is currently unexplored in India. The uniqueness of our study consisted of employing PLOs, who were members of the PLHIV support group, to follow up with participants. This peer group involvement has helped to minimise the attrition rates to 5.4%. 

Our study was initiated during the receding phase of the third wave of the pandemic, as a result of which, incidence of COVID-19 infection has declined and the majority of people might have attained natural immunity. This would have influenced the estimates of VE. Despite the efforts by the PLOs, there might be under-reporting of the symptomatic COVID-19 infections since the majority of our participants were daily wage workers and feared strict isolation if tested positive. The estimation of antibody titres to estimate the presence of prior COVID-19 infection would have validated our VE but it was beyond the scope of our study. Hence, knowledge regarding the prevention of infection is not available from the results of this study.

## 5. Conclusions

Our study estimated moderate effectiveness for full vaccination with either Covishield or Covaxin among PLHIV on HAART in Central Kerala. In comparison with Covishield, the vaccine effectiveness (VE) with two doses of Covaxin was estimated to be higher. A gap of 4 to 6 weeks between the two doses of the COVID-19 vaccine was found to have higher VE among PLHIV. The comparatively lower VE to the emerging COVID-19 variants among PLHIV reinforces the need to assiduously implement viral genomic surveillance to incorporate newer strains into the next generation of vaccines and warrants further studies quantifying the protective effect of the COVID vaccine.

## Figures and Tables

**Figure 1 viruses-15-02187-f001:**
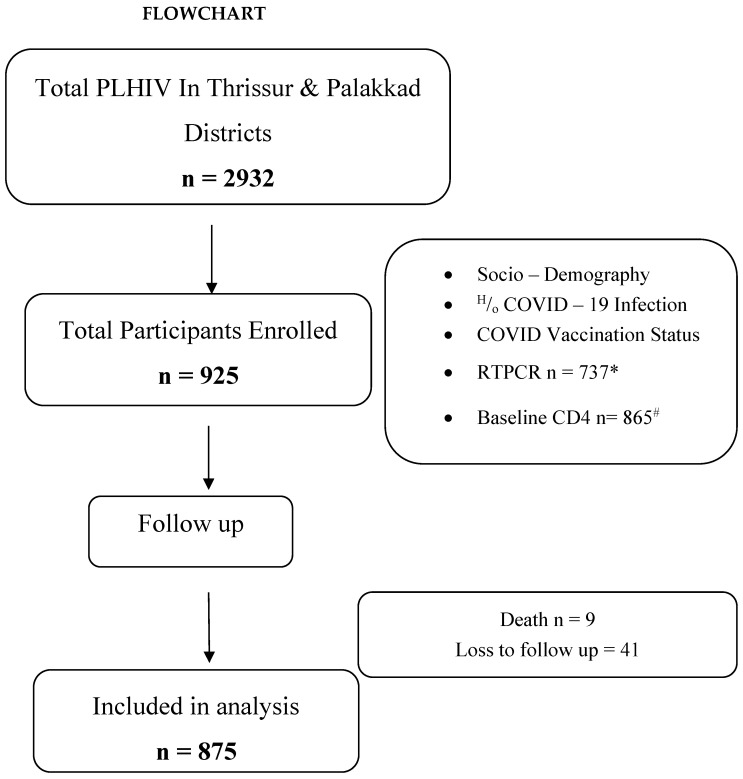
Flowchart showing the study procedure. * RTPCR was not conducted for 188 PLHIV as 116 did not give consent and 72 PLHIV had RTPCR performed within the past 10 days. # 60 participants did not consent for CD4 testing. @ Sputnik vaccinee was excluded from final analysis.

**Figure 2 viruses-15-02187-f002:**
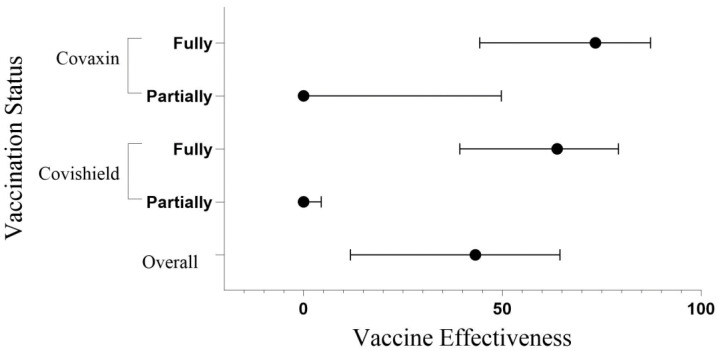
Vaccine effectiveness of first-generation vaccines among PLHIV.

**Figure 3 viruses-15-02187-f003:**
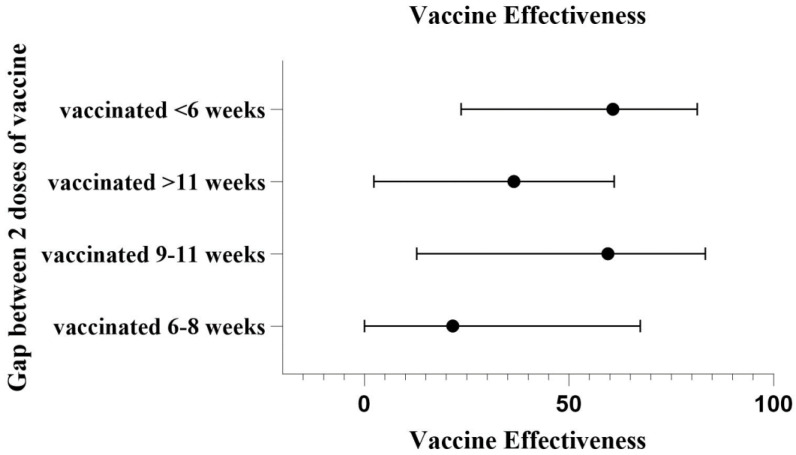
Vaccine Effectiveness according to the interval between the two doses.

**Figure 4 viruses-15-02187-f004:**
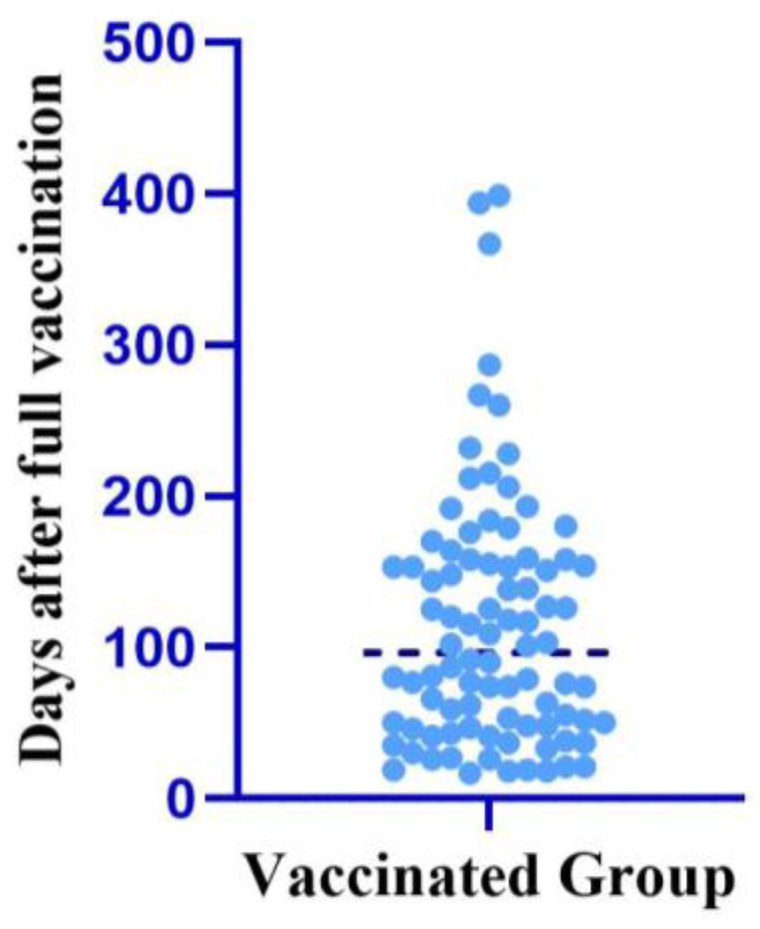
Mean duration to development of COVID-19 after vaccination.

**Table 1 viruses-15-02187-t001:** Sociodemographic profile.

Demographic Variables	Frequency (n = 925)	Percentage
Age		
≤30	78	8.4
31–40	145	15.7
41–50	358	38.7
51–60	257	27.8
61–70	78	8.4
>70	9	1.0
Mean ± SD	46.93 ± 11.00	
Sex		
Male	500	54.1
Female	424	45.8
Others	1	0.1
Marital Status		
Married	583	63.0
Unmarried	132	14.3
Widowed/Separated	194	21.0
Student	16	1.7
Education		
Primary	57	6.2
Secondary	684	73.9
Higher Secondary	133	14.4
Graduate/postgraduate	51	5.5
Occupation		
Employed	652	70.5
Unemployed	273	29.5

**Table 2 viruses-15-02187-t002:** Association between pre-existing comorbidities and COVID-19.

Variables	COVID-19 (n = 875)				RR (95% CI for RR)	*p* Value
	Positive		Negative			
	n	%	n	%		
Co morbidities						
Yes (n = 324)	75	23.1	249	76.9	1.250 (0.960–1.629)	0.099
No (n = 551)	102	18.5	449	81.5		
Diabetes Mellitus						
Yes (n = 131)	30	22.9	101	77.1	1.159 (0.820–1.638)	0.409
No (n = 744)	147	19.8	597	80.2		
Hypertension						
Yes (n = 73)	13	17.8	60	82.2	0.871 (0.522–1.452)	0.591
No (n = 802)	164	20.4	638	79.6		
Coronary artery disease						
Yes (n = 29)	7	24.1	22	75.9	1.201 (0.621–2.322)	0.594
No (n = 846)	170	20.1	676	79.9		
Stroke						
Yes (n = 12)	1	8.3	11	91.7	0.409 (0.062–2.681)	0.302
No (n = 863)	193	22.4	670	77.6		
Asthma						
Yes (n = 12)	3	25.0	9	75.0	1.240 (0.461–3.333)	0.958
No (n = 863)	174	20.2	689	79.8		
Dyslipidemia						
Yes (n = 66)	16	24.2	50	75.8	1.218 (0.778–1.907)	0.399
No (n = 809)	161	19.9	648	80.1		
Chronic Kidney Disease						
Yes (n = 7)	2	28.6	5	71.4	1.417 (0.436–4.606)	0.937
No (n = 868)	175	20.0	693	79.8		

**Table 3 viruses-15-02187-t003:** Vaccine effectiveness by type, dose, and interval between doses.

Variables	COVID-19				RR (95% CI for RR)	Vaccine Effectiveness	*p* Value	aRR (95% CI for RR)	Adjusted Vaccine Effectiveness	*p* Value
	Positive		Negative							
	n	%	n	%						
Vaccination Status										
Vaccinated (n = 819)	158	19.3	661	80.7	0.569 (0.384–0.841)	43.1 (15.9–61.6)	0.008	0.568 (0.355–0.882)	43.2 (11.8–64.5)	0.015
Unvaccinated (n = 56)	19	33.9	37	66.1	Reference			Reference		
Vaccination Status—Covishield										
Unvaccinated (n = 56)	19	33.9	37	66.1	Reference			Reference		
Partially Vaccinated (n = 149)	71	47.7	78	52.3	1.404 (0.939–2.100)	0 (0–6.1)	0.078	1.405 (0.956–1.867)	0 (0–4.4)	0.079
Fully Vaccinated (n = 500)	62	12.3	442	87.7	0.363 (0.235–0.559)	63.7 (44.1–76.5)	<0.001	0.362 (0.208–0.607)	63.8 (39.3–79.2)	<0.001
Vaccination Status—Covaxin										
Unvaccinated (n = 56)	19	33.9	37	66.1	Reference			Reference		
Partially Vaccinated (n = 19)	7	36.8	12	63.2	1.086 (0.543–2.171)	0 (0–45.7)	0.818	1.125 (0.503–1.915)	0 (0–49.7)	0.743
Fully Vaccinated (n = 150)	18	12.0	132	88.0	0.354 (0.201–0.623)	64.6 (37.7–79.9)	<0.001	0.266 (0.127–0.557)	73.4 (44.3–87.3)	<0.001
Gap between two doses										
Unvaccinated (n = 56)	19	33.9	37	66.1	Reference			Reference		
<6 weeks (n = 90)	12	13.3	78	86.7	0.393 (0.207–0.746)	60.7 (25.4–79.3)	0.003	0.393 (0.187–0.764)	60.7 (23.6–81.3)	0.004
6–8 weeks (n = 28)	7	25.0	21	75.0	0.737 (0.352–1.542)	26.3 (0–64.8)	0.404	0.786 (0.326–1.520)	21.6 (0–67.4)	0.529
9–11 weeks (n = 51)	7	13.7	44	86.3	0.405 (0.186–0.882)	59.5 (11.8–81.4)	0.015	0.405 (0.167–0.872)	59.5 (12.8–83.3)	0.018
>11 weeks (n = 546)	119	21.6	433	78.4	0.635 (0.427–0.947)	36.5 (5.3–57.3)	0.035	0.635 (0.390–0.977)	36.5 (2.3–61.0)	0.038

**Table 4 viruses-15-02187-t004:** CD4 count and vaccine effectiveness.

Vaccination Status	COVID-19 Status				RR (95% CI for RR)	Vaccine Effectiveness	*p* Value	aRR (95% CI for RR)	Adjusted Vaccine Effectiveness	*p* Value
	Yes		No							
	n	%	n	%						
**Overall**										
**CD4 (<350)**										
Vaccinated (n = 172)	31	18.0	141	82.0	0.793 (0.344–1.826)	20.7 (0–65.6)	0.593	0.836 (0.324–1.805)	16.4 (0–67.6)	0.684
Unvaccinated (n = 22)	5	22.7	17	77.3	Reference			Reference		
**CD4 (>350)**										
Vaccinated (n = 640)	125	19.5	515	80.5	0.466 (0.299–0.725)	53.4 (27.5–70.1)	0.003	0.466 (0.247–0.804)	53.4 (19.6–75.3)	0.004
Unvaccinated (n = 31)	13	41.9	18	58.1	Reference			Reference		
**Covishield**										
**CD4 (<350)**										
Vaccinated (n = 134)	26	19.4	108	80.6	0.854 (0.367–1.986)	14.6 (0–63.3)	0.717	0.923 (0.352–1.967)	7.7 (0–64.8)	0.856
Unvaccinated (n = 22)	5	22.7	17	77.3	Reference			Reference		
**CD4 (>350)**										
Vaccinated (n = 510)	105	20.6	405	79.4	0.491 (0.314–0.768)	50.9 (23.2–68.6)	0.005	0.491 (0.261–0.842)	50.9 (15.8–73.9)	0.007
Unvaccinated (n = 31)	13	41.9	18	58.1	Reference			Reference		
**Covaxin**										
**CD4 (<350)**										
Vaccinated (n = 38)	5	13.2	33	86.8	0.579 (0.188–1.780)	42.1 (0–81.2)	0.338	0.579 (0.163–1.645)	42.1 (0–83.7)	0.343
Unvaccinated (n = 22)	5	22.7	17	77.3	Reference			Reference		
**CD4 (>350)**										
Vaccinated (n = 129)	20	15.5	109	84.5	0.370 (0.207–0.659)	63 (34.1–79.3)	0.001	0.369 (0.172–0.72)	63.1 (28–82.8)	0.002
Unvaccinated (n = 31)	13	41.9	18	58.1	Reference			Reference		

## Data Availability

Data are unavailable due to privacy and ethical restrictions.

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
