# Peer review of "Real-World Effectiveness of COVID-19 Vaccine and Identification of SARS-CoV-2 Variants among People Living with HIV on Highly Active Antiretroviral Therapy in Central Kerala of India—An Ambi-Directional Cohort Study"

_viruses, 2023, doi:10.3390/v15112187_

Round 1

Reviewer 1 Report

Well written paper and relevant research question.

Minor comments

line 35-36 ".. CD4 count .." should read "... baseline CD4 count .. "

table 1

Label "3" for sex: "Transgender", as in text (line 153)

Label "Degree and above" not clear

____

Minor revision of English required (use of definite/indefinite article).

Author Response

Thank you for the review.

Following are the reply to the reviewer's comments

Abstract: Line 35 -36 : Correction done to baseline CD4 count as suggested

Table 1: Line 153 : Gender category 3 modified to "Others"

Table 1: Education : correction done to graduate/postgraduate

English language revision noted and shall be done 

Thank you

Reviewer 2 Report

Overall comments

COVID-19 has been a major health challenge of recent years, but vaccines have made a significant contributions to blunting the effects of the COVID-19 pandemic.. HIV is a long-standing health challenge around the world, including in India. The immunodeficiencies that accompany HIV infection render patients more susceptible to severe COVID-19 disease and make HIV less able to mount an effective immune response following vaccination. The paper presents useful data describing the vaccine efficacy of two COVID-19 vaccines widely used in India, AZD1222/Covishield and BBV152/Co- 56 vaxin, in People Living with HIV (PLHIV) in Kerala.

The study showed that this group of PLHIV had an admirably high rate of COVID-19 vaccination, and a good rate of vaccine effectiveness, consistent with other studies in other populations vaccinated with the vaccines used.

Overall, the manuscript makes a useful contribution to the understanding of vaccine effectiveness of the vaccines used in India in the highly vulnerable PLHIV population and shows that PLHIV with good viral load suppression are well protected against COVID-19 by vaccines in use in Kerala.

Major

The writing must be improved. There are many grammatical and usage errors. Many articles are omitted or used incorrectly. Abbreviations are used inconsistently

Minor

All abbreviations, such as PLHIV should be defined on first usage, including in the abstract.

Check that author names are correct, some appear to have initials following surnames, while other authors seem to have initials precede surnames.

Background

Appropriate references should be supplied for the vaccines under study.

Introduction

More information should be provided about the state of the COVID-19 pandemic in Kerala during the time of the study.

Methods

Presumably, a very large fraction of the patients enrolled in the study had personal phones. Is any data available for this?

More details should be given concerning the COVID-19 diagnostic tests used.

How were HIV viral loads and CD4 counts determined?

Is any data available on the duration of viral load suppression? This would be helpful in assessing the consistency of antiretroviral usage.

Is any information available on the mode of HIV infection of the participants?

Results

The age distribution of the study participants seems to skew old for India. Is that correct and if so, are there any explanations?

The manuscript says that 82,9% of participants were being treated with tenofovir, lamivudine, dolutegravir (TLD). Please provide treatment information for the other 17,1%. What fraction of participants were no being treated with antiretroviral therapy at all?

The manuscript needs to say how many participants had very low CD4 counts, <100, <50. The manuscript needs to estimate vaccine effectiveness for these cohorts with lower CD4 counts.

What are the lower limits of detection of the viral load assays?

The manuscript needs to analyze vaccine effectiveness as a function of the antiretroviral therapy regimen used (even though a large majority received DLT), and as a function of viral load suppression.

Conclusions/Discussion

These are reasonable given the data.

The Discussion section should note, among the limitations of the study, that no serologic testing was done on the participants prior to entry into the study, so that no knowledge concerning prior infection with SARS-CoV-2 was available, which could have a significant impact on clinical vaccine effectiveness.

Also among limitations, the Discussion section should mention that no knowledge is available from this study concerning the prevention of infection, only prevention of clinically notable disease in the vaccine effectiveness calculations.

Figures and Tables

Reasonable, but figures for the additional analyses mentioned above should be included.

Extensive improvements needed.

Author Response

Major comments :

The grammatical errors and abbreviations will be corrected as per the suggestion

Minor:

All abbreviations expanded on first use including the abstract. Author names rechecked and they are correct. Few authors have their initials at the beginning of the name

Background: References for the vaccines under study included in the introduction section of the manuscript

Introduction: Status of COVID-19 pandemic during the study period was explained in the discussion. As per suggestion this data has been included in the introduction section.

Methods:

  1. Patients enrolled in the study were followed up using telephonic interviews and all the subjects had their personal mobile phones. There were 41 loss to follow up subjects in the study as mentioned in our flowchart. These included subjects who were not responding to the weekly telephonic calls by our Participant Liason Officers.
  2. The COVID-19 testing done in this study included only RTPCR tests done at laboratories designated by Indian Council of Medical Research(ICMR). This has been mentioned in  methodology - Lines 123, 124
  3. CD4 counts were detected by flow cytometry technique and HIV Viral load by nucleic acid amplification test using Roche COBAST TaqMan HIV-1 and Abbott Real Time HIV-1
  4. Viral load testing is done yearly once for the subjects reporting at the ART Centre. Hence the duration of VL suppression in this study would be one year. This data has been added to the results section
  5. Mode of HIV infection was not a part of our questionnaire and hence this data is not available 

Results:

  1. Age distribution being skewed to older group could be because the younger group may be more concerned about revealing their identity and details for research. 
  2. Other than TLD regimen, 17.1% included subjects with TLE and also those using second line ART medications. Since our data collection was done at the ART centre where the subjects came for data collection, all the individuals were currently on medications
  3. Participants with CD4 count less than 100 were 21(2.5%) and less than 50 were 4(0.5%). Vaccine effectiveness of the cohorts with CD4 count less than 100 was calculated as per suggestion and estimated to be 76.5% (95% CI: 24 - 92.7%) p=0.022. This data has been added to results 
  4. Viral load suppression was reported an TND (Target not detected) in the results obtained. This indicated they had less than 20 copies per millilitre of blood.
  5. VE as a function of ART and based of viral load has been done seperately and added to the results section

Discussion:

  1. The limitation of not doing the serology/antibody titre has been briefly mentioned in the discussion section line 329-330. It has been made more clear as per suggestion
  2. Only prevention of clinically noticed and tested disease and not prevention of infection is available from the VE calculations in this study. This will be added in the discussion section as per suggestion.

Figures and tables: additional analyses will be added to the manuscript and tables if required.

Round 2

Reviewer 2 Report

The authors have made some reasonable responses to the reviewers' comments, but there remain usage problems and typos. These can be handled at the copy editing stage.

There are still usage problems and inconsistent phrasing. Article usage often follows colloquial Indian usage and not standard English.